# Hydrodynamic performance of Ordovician archaeostracan carapaces

Stephen Pates[1,2,3]*, Yuan Xue[3]

1 Centre for Ecology and Conservation, University of Exeter, Penryn, Cornwall, United Kingdom,
2 Department of Zoology, University of Cambridge, Cambridge, United Kingdom, 3 Homerton College,
University of Cambridge, Cambridge, United Kingdom

* s.pates@exeter.ac.uk

## Abstract

The diversification of macroscopic pelagic arthropods such as caryocaridid archaeostracans was a crucial aspect of the Great Ordovician Biodiversification Event, and the plankton revolution. A pelagic mode of life has been inferred for caryocaridids from their common presence in black graptolitic shales alongside carapace morphologies that appear streamlined. However, the hydrodynamic performance within the group and comparisons with other archaeostracans were lacking. Here we use a computational fluid dynamics approach to quantify the hydrodynamic performance of caryocaridids, and other early Palaeozoic archaeostracans including *Arenosicaris inflata* and Ordovician ceratiocaridids. We show that streamlining of the carapace was an important factor facilitating a pelagic mode of life in caryocaridids, in reducing the drag coefficient and facilitating a broader range of lift coefficients at different angles of attack. However, comparable hydrodynamic performance is also recovered for some ceratiocaridids. This suggests that alongside carapace streamlining, adaptations to appendages and thinning of the carapace were also important for a pelagic mode of life in Ordovician caryocaridids.

## Introduction

Arthropods are a major component of the pelagic realm, functioning as intermediaries between primary producers and tertiary consumers in marine food webs and connecting the pelagic and benthic realms [1, 2]. In doing so, they play key roles as ecosystem engineers, repackaging nutrients and organic matter as faecal pellets that are transported vertically through the biological pump [2–4] and biologically mixing the water column [5, 6]. Indeed the early Palaeozoic expansion and diversification of arthropods into the water column was one of a series of evolutionary and environmental feedbacks that transformed the oceans from a stratified, micro-organism dominated, water column with oxygen available only near the surface, to a well-ventilated one where an animal-dominated biological pump links benthic and pelagic realms [1, 6–12].

While pelagic arthropods are known from the early Cambrian [1, 7, 8, 11] the synchronous radiation of phytoplankton, microscopic zooplankton and macroscopic pelagic animals with

**Data Availability Statement:** Data have been uploaded to an OSF project. Doi: 10.17605/OSF.IO/JV32T

**Funding:** SP received funding through a Herchel Smith Postdoctoral Fellowship (no number). https://www.herchelsmith.cam.ac.uk/postdoctoral-

fellowships The sponsors had no role in study design, data collection and analysis, decision to publish, or preparation of the manuscript.

**Competing interests:** The authors have declared that no competing interests exist.

benthic suspension feeders as part of the plankton revolution in the late Cambrian to early Ordovician led to a further ecological revolution in early Palaeozoic oceans [13, 14]. Arthropods were critical at a number of different size and trophic classes within this ecological revolution, from microscopic planktonic trilobite larvae [15] to large meter scale filter feeders [16, 17].

Ordovician macroscopic pelagic arthropods (size range 20–200 mm) [1] include trilobites [18, 19] and caryocaridid archaeostracans [20, 21]. The morphological characteristics and cosmopolitan distribution of caryocaridids, including their carapace shape and common presence in outer shelf or slope settings alongside graptolites in black shales, supports a pelagic life mode at the fringes of continental margins for this group [20–23].

In this study we explore the hydrodynamic performance (drag and lift) of caryocaridid carapaces, contrasting them with archaeostracan relatives including the late Cambrian benthic *Arenosicaris* [24] and Ordovician ceratiocaridids, which are generally considered pelagic [25] or nektobenthic [26]. Thus we determine the role of carapace shape in facilitating a pelagic mode of life in Ordovician caryocaridids.

## Materials and methods

### Taxonomic sampling

Sixteen taxa were chosen, sampled from terminals in the phylogenetic analysis of Liu et al. [21]. Taxa were chosen ensuring a broad range across the phylogeny, including at least one representative from all clades of at least two taxa and at least one representative of each genus (Fig 1).

### Carapace outlines

Two dimensional outlines of 15 species of Ordovician and one late Cambrian archaeostracans were generated from reconstructions in the literature (14 from Ref [21], and one each from Refs [24, 27]) using Inkscape 1.0. For the literature source and figure used for each outline, see the OSF project [28]. Idealised reconstructions of taxa–rather than specific specimens–were chosen for analysis. Factors including intraspecific variation, ontogeny, dimorphisms, as well as post-mortem deformation, are expected to introduce differences in the shapes of carapaces of individual species–as previously noted for a range of Cambrian and Ordovician non-biomineralizing total group euarthropods [29–36]. Idealised reconstructions of taxa from literature sources are likely sufficient to inform on the hydrodynamic implications of intraspecific differences in archaeostracan carapace morphology. This is because the introduction of any slight inaccuracies in idealised carapace outlines are not as large as the interspecific morphological differences they depict. For this study, the carapace reconstructions of *Ceratiocaris angusta* and *Janviericaris raymondi* are ultimately based off incomplete material. For the former, the incomplete portion represents a small part of the antero-dorsal margin, whereas for the latter it represents much of the anterior of the carapace. Thus the results for these two taxa, in particular *J. raymondi*, will have a higher uncertainty–resulting from incomplete morphological data–than the other taxa in the study. Future discoveries of more complete material will allow refinement of the carapace outlines, and, if necessary, additional simulations to be performed.

Outlines of individual taxa were then scaled, centered and aligned, before being sampled to the same resolution (64 points) and converted to a text file containing coordinates of the 64 points that could be read directly into ANSYS DesignModeler (Ansys Academic 2021 Release R2; Ansys Academic 2023 Release R2), using R [37] Momocs package [38] and a function from [11]. Some shapes had rows of small posterior spines which created spaces too small for the meshing program to work, while for others sampling the outline at 64 points led to the anterior spine not being captured with sufficient detail. For these shapes, outlines were modified to remove small posterior spines (but leaving in the dorsalmost and ventralmost spines)

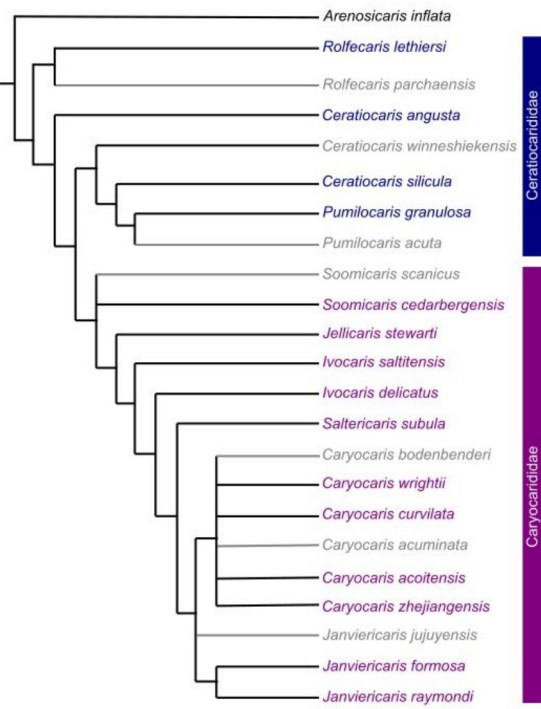

**Fig 1. Cladogram of Ordovician archaeostracans.** Adapted from Liu et al. [21, Fig 2]. Taxa not selected for computational fluid dynamics analyses in grey.

and sampled at 256 points in order to capture anterior spine morphology. All R code and outlines are provided in the accompanying OSF project [28].

## Benefits and limitations of a 2D approach

The use of two-dimensional outlines facilitates comparison of a broad range of taxa, including some known only from a few specimens. In contrast, a three-dimensional approach would have required taxa where specimens are preserved in a range of orientations, thus limiting the range of taxa available for the analyses. By focusing on a single aspect–the outline shape of the carapace–the impact of this can be isolated from other aspects, such as the angle between the margins of the carapace and the thickness of the carapace. This also represents a limitation, as the two-dimensional nature of the analyses does not allow these other factors, which are likely also of hydrodynamic importance, to be assessed. When considering the fossil data available, interspecific differences in the carapace thickness and angle between the margins of the carapace are more challenging to ascertain. Thus, a focus on interspecific differences in carapace shape allow this study to interrogate the implications of interspecific differences in morphology that can be observed in the fossil record.

## Computational fluid dynamics

Following import into ANSYS DesignModeler, models were scaled to a length of 25mm. The geometry, meshing and model set-up of [11] were used as this has been validated and verified, and assessed the hydrodynamic performance of carapace valves of similar morphology and size (Fig 2). This meshing set up comprises an edge sizing control applied to the margin of the carapace, a sphere of influence with a fine element size applied to the area immediately around

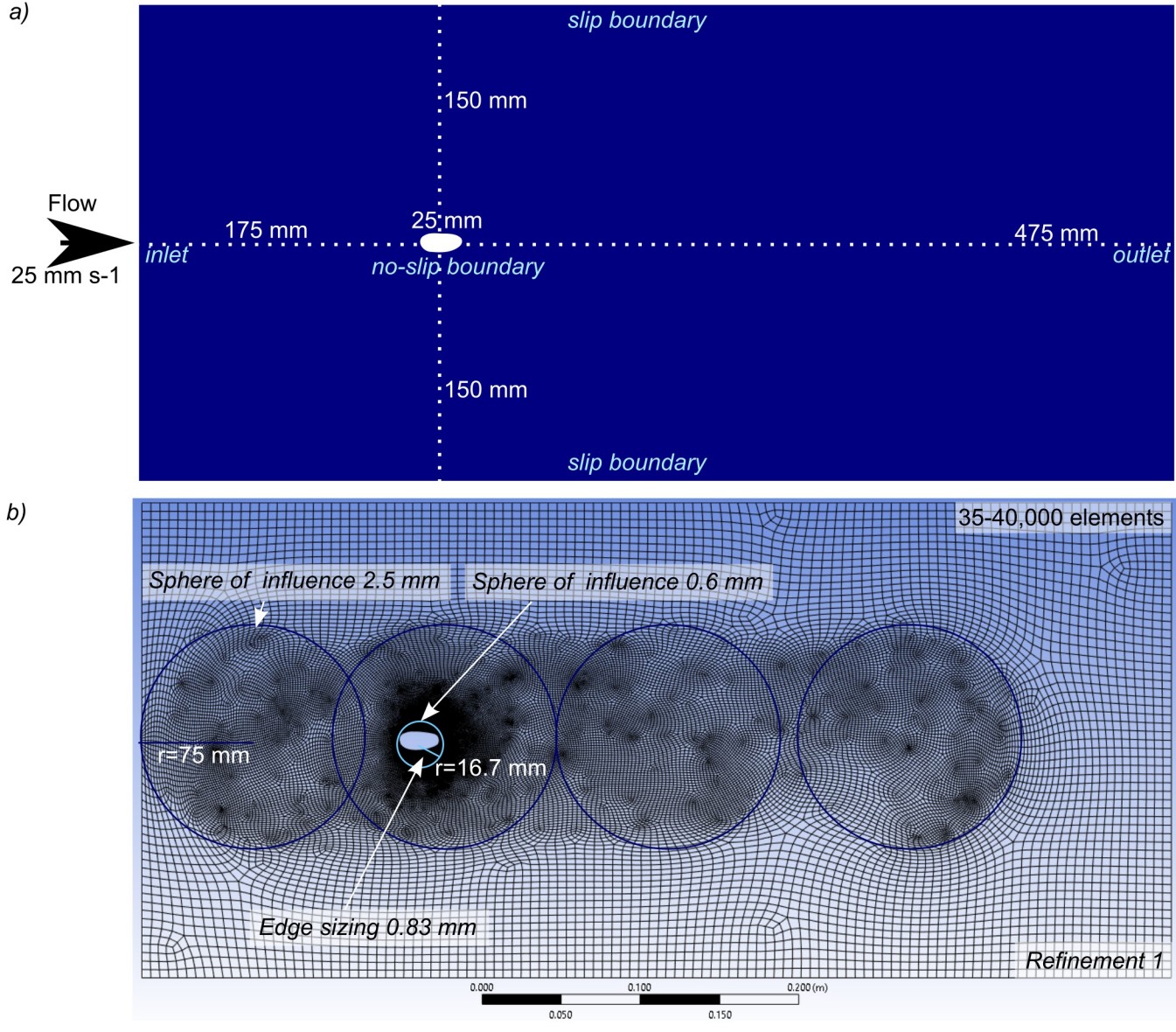

**Fig 2. Computational fluid dynamics design and set-up.** Geometry and boundary conditions for simulations (a) and meshing set up (b) used in this study. Element sizes given next to meshing controls. Whole domain subjected to Refinement 1 (Ansys Meshing; Ansys Academic 2021 release R2 and 2023 release R2).

the carapace, and four spheres of influence with a moderate element size applied to anterior to and posterior to the carapace (Fig 2). A refinement control was applied to the whole region, giving approximately 35–40,000 elements for the meshes used in the simulations. A more refined mesh (c. 80–85,000 elements) and less refined mesh (c. 9,000 elements) were used to test whether the chosen mesh was suitably refined. Archived ANSYS project files are available in the OSF project [28].

CFD simulations were conducted using the steady-state laminar solver in ANSYS Fluent (Ansys Academic, Release 2021 R2 and Release 2023 R2). Forces and coefficients of drag and lift were calculated for a flow speed of one carapace length (25 mm) per second. Modern crustaceans of a similar size to archaeostracans have been reported to swim between 0.5–3 body lengths per second [39]. Simulations of flow around the carapace of the stem group

euarthropod *Isoxys* that considered multiple velocities (from 0.75–1.18 body lengths per second) recovered the same comparative relationships (drag, lift) between the taxa [11], and so only one flow speed was used for this study. The carapace length (25 mm) was taken as the reference area for calculation of coefficients, giving a Reynolds number for these simulations of 340. A single carapace length of 25mm facilitates size-independent comparisons of hydrodynamic performance, excluding the impact of size differences from the interpretation. This standardized carapace length is representative of most archaeostracans considered in this study. *Arenosicaris* carapaces range from 28.6–36.2 mm in length (mean 32.6 mm) [24], the carapace of the first figured specimen of *Ceratiocaris angusta* is 25 mm in length [40], while those of *Ce. silicula* are smaller, c. 10–15 mm [25]. Caryocaridids are generally between 10–30 mm in length [21].

Solutions were considered converged when residuals were less than $10^{-6}$. Simulations were conducted at a range of angles of attack at 2 degree increments, to find the stall angle and minimum drag generated by each carapace outline. If a solution did not converge (residuals remained greater than $10^{-6}$ and sinusoidal drag and lift forces generated across iterations) the flow was considered to be unsteady, and no drag or lift values were recorded. A further simulation at 1 degree less angle of attack was then conducted in many cases, to find the stall angle as accurately as possible. Simulations were performed at negative angles of attack until flow was unsteady or the drag coefficient was equal to or greater than that at the stall angle.

## Coefficients and calculations

Coefficients of drag and lift ($C_d$, $C_l$) were calculated for each converged simulation. Coefficients of drag and lift are proportional to the force (either drag or lift, $F_d$ and $F_l$ respectively), inverse square of swimming speed (u) and the inverse of the reference length (A) (Eqs 1 and 2; $\rho$ is the density of the fluid).

$$C_d = \frac{2F_d}{\rho u A} \tag{1}$$

$$C_l = \frac{2F_l}{\rho u A} \tag{2}$$

Drag:lift ratios were calculated by dividing the drag coefficient by the lift coefficient at the same angle of attack.

## Results

### Mesh sensitivity of CFD simulations

In mesh sensitivity analyses for the computational fluid dynamics simulations, results were considered mesh independent if the values recovered for drag and lift coefficients for a given shape and angle of attack were less than 2% different when compared to the same coefficients recovered from the more refined mesh. A refinement 1 mesh (35–40,000 elements) was chosen for all carapaces, as in all cases these results were within the mesh independent threshold (2%) compared to the refinement 2 mesh (c. 80–85,000 elements). In some cases, results from the starting mesh were less than 2% different to the refinement 1 mesh, however a refinement 1 mesh was still used in these calculations as it was not significantly more computationally expensive, and so approximately the same number of elements were used in simulations for all taxa. Full results of the mesh sensitivity simulations are provided in the OSF project [28]. A

difference of 2% in drag and lift coefficients is far less than the observed differences between the taxa (results below), supporting the use of this value as the mesh independence threshold.

## Drag and lift coefficients

A total of 265 simulations were run, 48 of which were for mesh sensitivity analyses (three per taxon). Sixteen of these (one per taxon for the chosen mesh refinement) were combined with 217 further simulations to provide results for drag and lift coefficients and different angles of attack for the 16 taxa in the study (Table 1).

Archaeostracan carapaces display a broad range of drag and lift coefficients (Fig 3). The carapaces with the highest drag coefficients belonged to *Arenosicaris inflata* and two ceratiocaridids (*Pumilocaris granulosa* and *Rolfecaris lethiersi*). Drag coefficients were lower in all caryocaridids and the two *Ceratiocaris* species (*C. angusta* and *C. silicula*). All caryocaridids displayed broad ranges of lift coefficients including both negative and positive values, with the exception of *Janviericaris formosa* (positive values only). Lift coefficient ranges were broadest for *Caryocaris acoitensis*, *Ca. zhejiangensis* and *Saltericaris subula*.

## Pressure and velocity contours

Carapace shape affected flow, which can be visualised by comparing the pressure and velocity contours around *Arenosicaris inflata* with a caryocaridid and ceratiocaridid (Figs 4 and 5). The larger area of high pressure at the anterior of the *Arenosicaris inflata* carapace, compared to *Ca. acoitensis* and *Ce. silicula* (Fig 4), is likely responsible for the higher drag. Visualisation of the velocity contours for the same carapaces reveals a broader and more elongate area of low velocity flow posterior to the carapace for *A. inflata* than for *Ca. acoitensis* and *Ce. silicula* (Fig 5).

**Table 1. Number of CFD simulations for each taxon.**

| Taxon | Number of simulations |
|---|---|
| *Arenosicaris inflata* | 10 |
| *Caryocaris acoitensis* | 18 |
| *Caryocaris curvilata* | 14 |
| *Caryocaris wrightii* | 16 |
| *Caryocaris zhejiangensis* | 18 |
| *Ceratiocaris angusta* | 16 |
| *Ceratiocaris silicula* | 18 |
| *Ivocaris delicata* | 12 |
| *Ivocaris saltitensis* | 14 |
| *Janviercaris formosa* | 12 |
| *Janviercaris raymondi* | 19 |
| *Jellicaris stewartia* | 19 |
| *Pumilocaris granulosa* | 11 |
| *Rolfecaris lethiersi* | 3 |
| *Saltericaris subula* | 18 |
| *Soomicaris cedabergensis* | 15 |

Number of reported simulations does not include ones solely run for mesh refinement. Three mesh refinement simulations were run for each taxon, one of which is included in the results plotted as Fig 3.

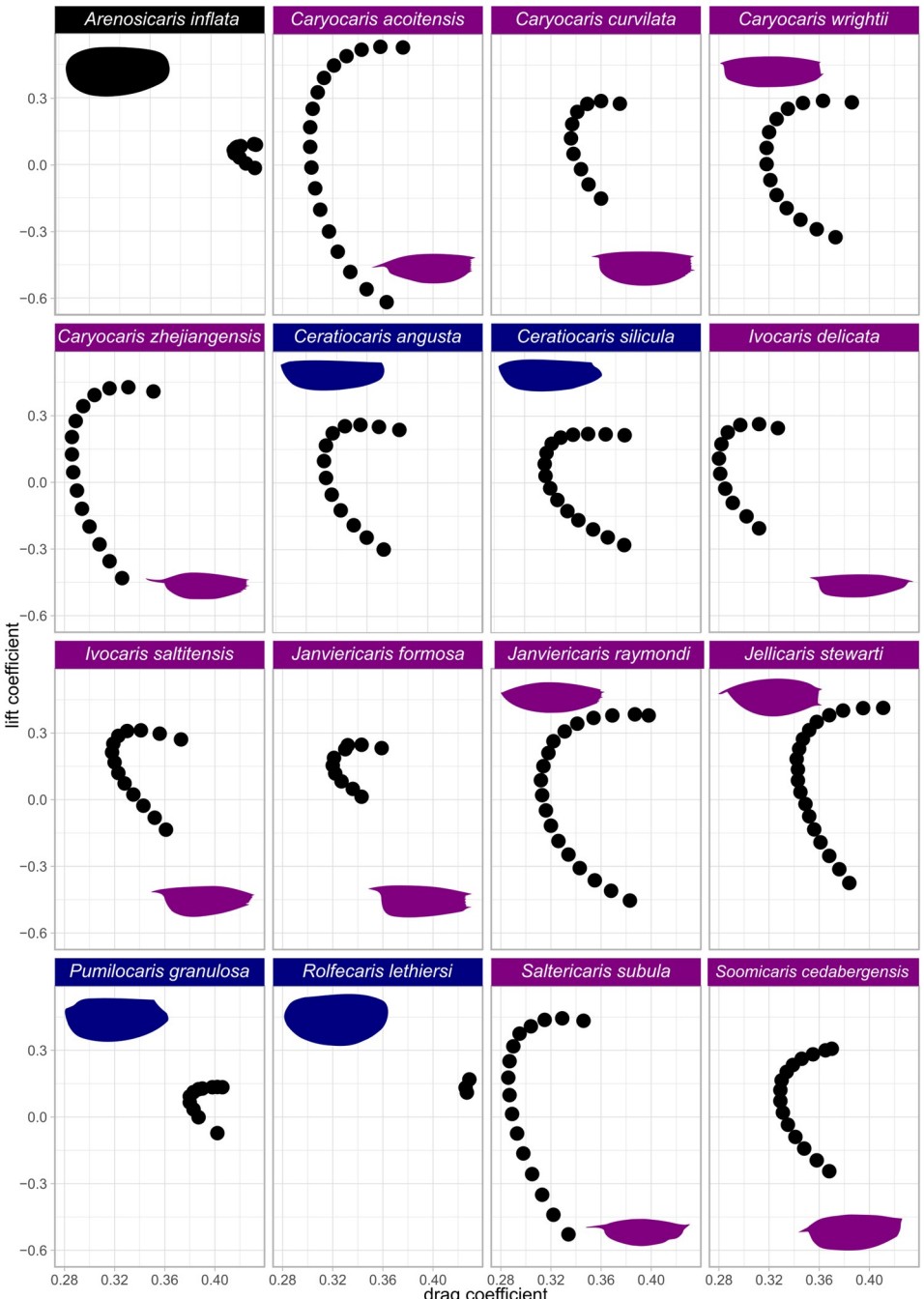

**Fig 3. Results of the computational fluid dynamics simulations.** Drag polars for all taxa in this study, overlain by carapace outline used in computational fluid dynamics simulation. Each point represents results (drag and lift coefficients) of a single simulation at Reynolds number of 340. Full results including mesh refinement provided in OSF project, alongside ANSYS project files [28].

## Phylogenetic context

Plots of minimum drag coefficient against highest lift:drag ratio demonstrate a likely relationship between these two metrics of hydrodynamic performance (Fig 6). Carapaces of taxa at a greater tree path distance (number of nodes used as a proxy) from *Arenosicaris inflata* (Fig 1)

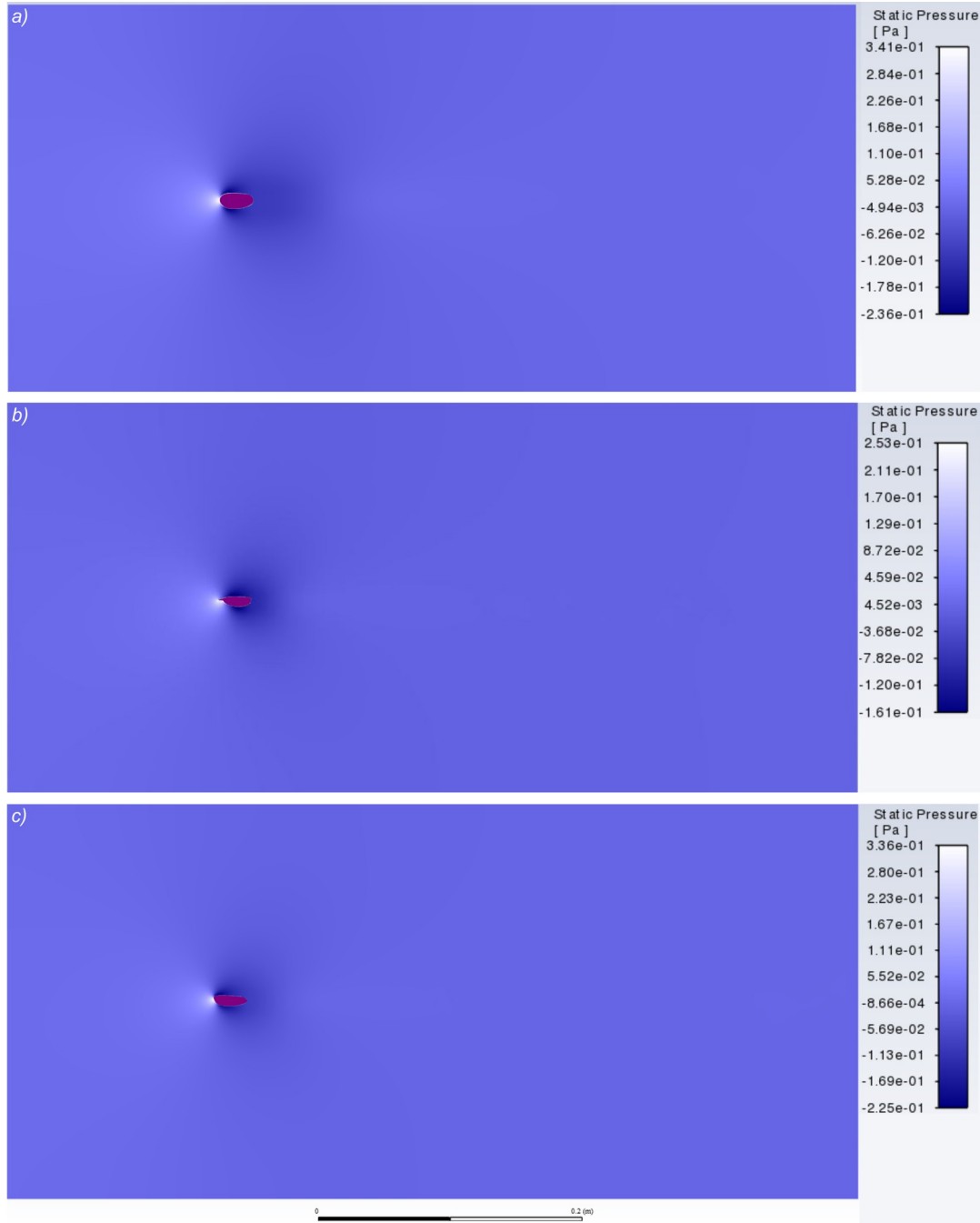

**Fig 4. Contours of static pressure.** Around carapaces of *Arenosicaris inflata* (a), *Caryocaris acoitensis* (b), and *Ceratiocaris silicula* (c). Angle of attack for all three set to 0˚. Visualized using ANSYS Fluent. Length of carapaces 25 mm.

are broadly more hydrodynamic (higher lift:drag ratio, lower minimum drag coefficient) than those fewer nodes away. The sample size is too small to statistically test for correlation between tree path distance from *A. inflata* (number of nodes used as a proxy) and highest lift:drag ratio, or minimum drag coefficient.

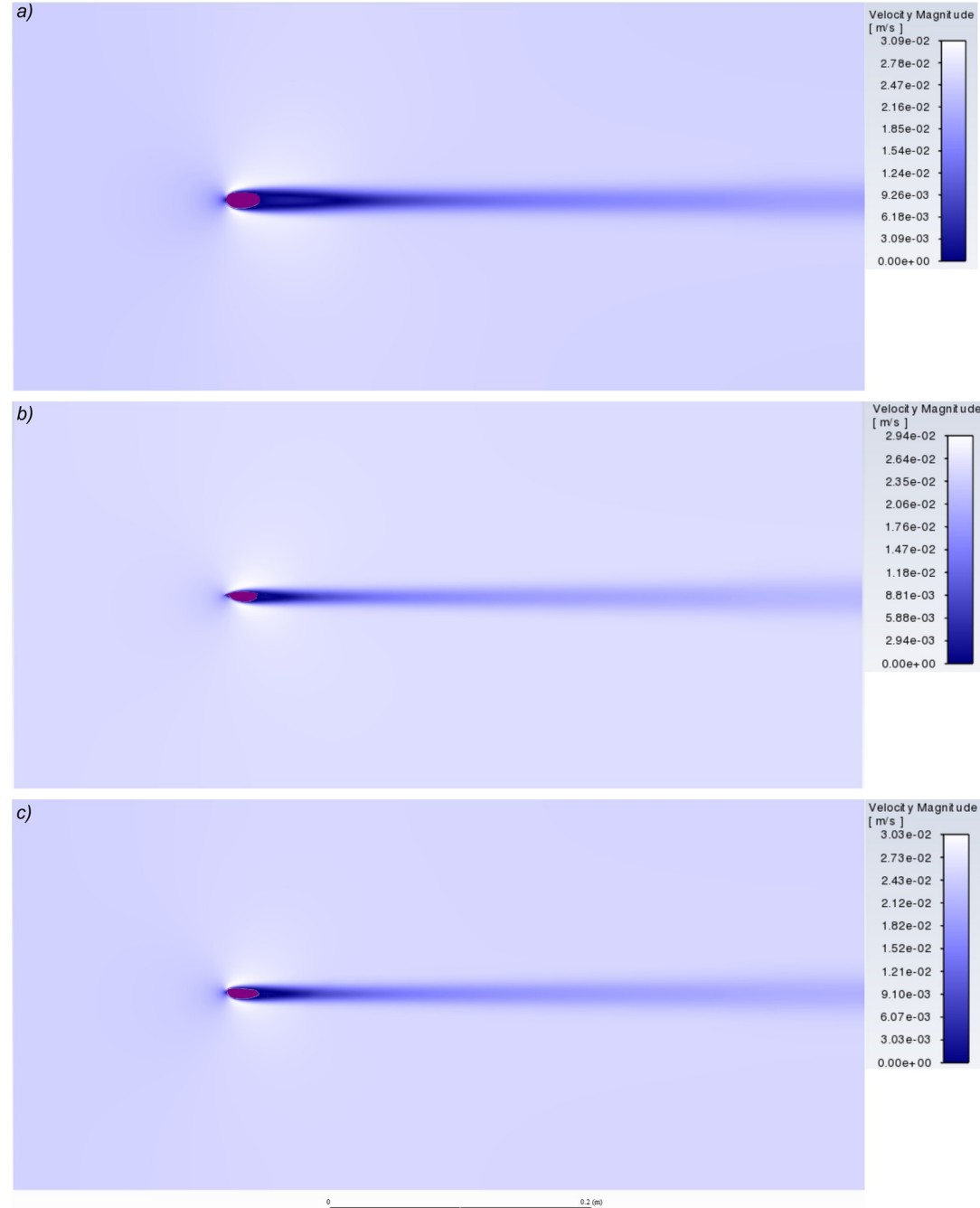

**Fig 5. Contours of velocity magnitude.** Around carapaces of *Arenosicaris inflata* (a), *Caryocaris acoitensis* (b), and *Ceratiocaris silicula* (c). Angle of attack for all three set to 0˚. Visualized using ANSYS Fluent. Length of carapaces 25 mm.

## Discussion

The carapaces of caryocaridids outperformed *Arenosicaris inflata* and two of four ceratiocaridids considered in terms of minimum drag coefficient, range of lift coefficients and lift:drag ratios. This indicates that improved hydrodynamic performance of the carapace was an important factor in facilitating a pelagic mode of life in Ordovician caryocaridids, providing

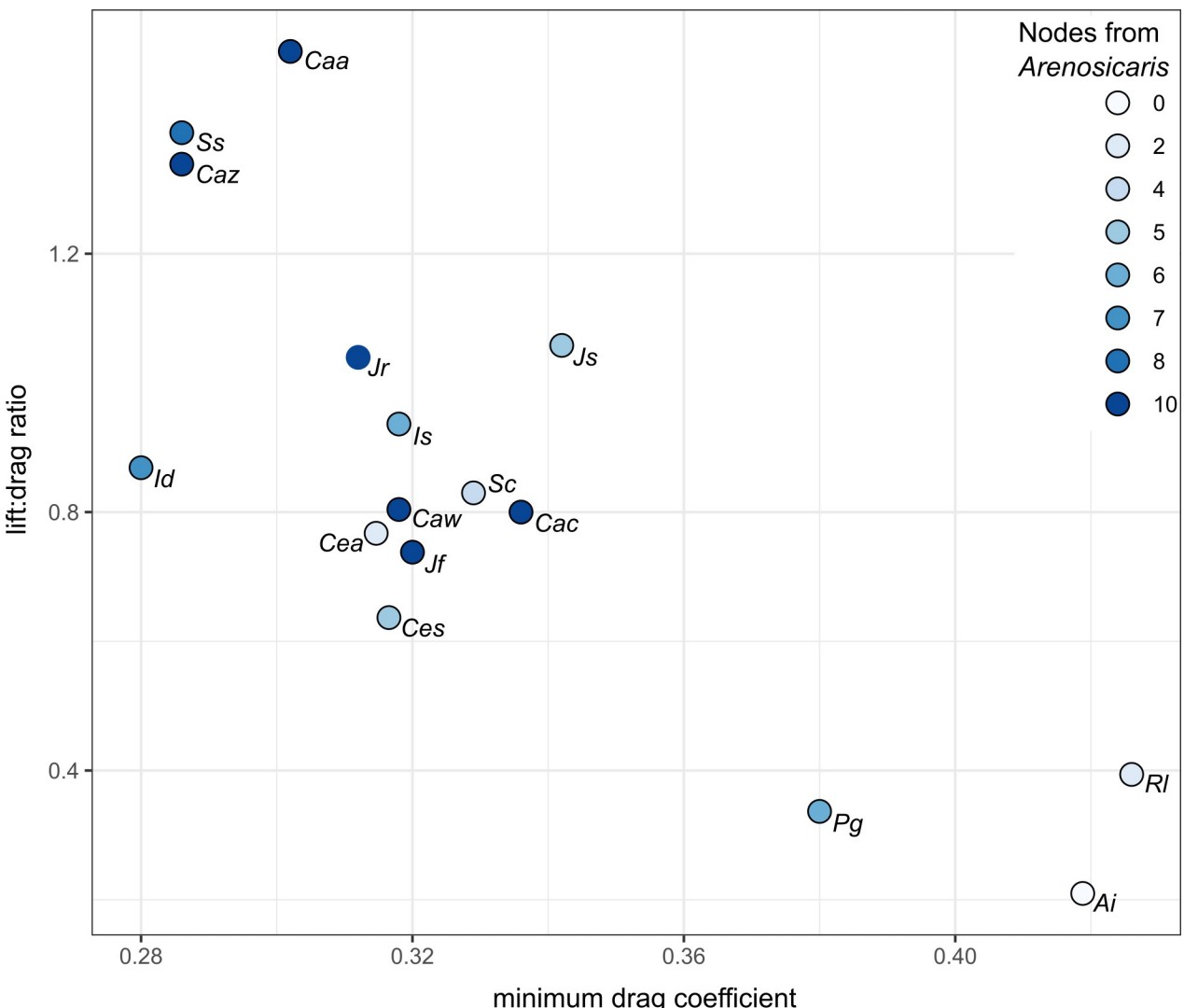

**Fig 6. Plot of minimum drag coefficient against highest lift:drag ratio for all carapaces analysed in this study.** Color of points indicates number of nodes away from *Arenosicaris inflata* (according to cladogram presented in **Fig 1**, which is adapted from Liu et al. [21, Fig 2]. Abbreviations: *Ai, Arenosicaris inflata; Caa, Caryocaris acoitensis; Cac, Caryocaris curvilata; Caw, Caryocaris wrighti; Caz, Caryocaris zhejiangensis; Cea, Ceratiocaris angusta; Ces, Ceratiocaris silicula; Id, Ivocaris delicata; Is, Ivocaris saltitensis; Jf, Janviericaris formosa; Jr, Janviericaris raymondi; Js, Jellicaris stewarti; Pg, Pumilocaris granulosa; Rl, Rolfecaris lethiersi; Ss, Saltericaris subula; Sc, Soomicaris cedabergensis.*

quantitative support for qualitative assessments by previous workers [20, 21]. Lower drag coefficients were facilitated through a high length:depth ratio of carapaces. The role of the anterior spine is less clear. The lowest drag coefficients were obtained for taxa with a short anterior spine (e.g. *Ca. zhejiangensis, I. delicata, S. subula*), but these taxa also display elongate and narrow carapaces. Low drag coefficients were also obtained for taxa lacking an anterior spine (*Ce. angusta, Ce.silicula*) indicating that carapace shape is more important (Fig 3). Taxa with the highest drag coefficients lacked anterior spines and had broader carapaces with lower length: depth ratios relative to other archaeostracans (*Arenosicaris inflata, Pumilocaris granulosa, Rolfecaris lethiersi*).

*Janviericaris formosa* is notable amongst the caryocaridids for its relatively limited range of lift coefficients, which are all positive values (Fig 3). These results indicate that the carapace

was not well adapted for a pelagic mode of life with regular vertical movements in the water column. Most caryocaridids (and the two *Ceratiocaris* species simulated) have carapaces that generate negative or positive lift, depending on the orientation of the carapace in the flow (Fig 3). Thus, these taxa could have moved vertically upwards and downwards by changing their orientation. In contrast, descent in the water column by *J. formosa* could only have been achieved through passive sinking (assuming a negative buoyancy). Notably the maximum lift coefficient of *J. formosa* is also lower than other caryocaridids, although it does display a similar lift:drag ratio to *Caryocaris curvilata*, *Ca. wrighti*, *Soomicaris cedabergensis* and the two *Ceratiocaris* species (Fig 6). The limits on a pelagic mode of life *for J. formosa* is corroborated by the geological evidence, where this taxon is found only in shelf deposits [41], contrasting to other caryocaridids which are found in slope and/or basin deposits [41]. *Soomicaris cedabergensis*, interpreted as an epipelagic representative of the group [42], displays comparable hydrodynamic properties to *Caryocaris* taxa interpreted as mesopelagic (Fig 3).

The hydrodynamic performance of the carapaces of the two sampled *Ceratiocaris* species, *C. angusta* and *C. silicula* are broadly comparable to those of most caryocaridids (Figs 3 and 6). Notably a pelagic lifestyle for *C. silicula* has been previously suggested on account of its narrow and elongate carapace [25], while *C. angusta* was initially described as a caryocaridid partly on account of its carapace proportions [40]. Collette and Hagadorn [25] posited that the superficial similarities in carapace shape, which lead to the similar drag and lift coefficients from the simulations, could have resulted from ecological convergence to a pelagic mode of life or from a shared ancestral form. In the context of the phylogenetic results of Liu et al. [21] (Fig 1), and the hydrodynamic simulations presented herein (Fig 6), a slender hydrodynamic carapace likely originated at the common ancestor of all Ordovician archaeostracans, and was secondarily lost in *Pumilocaris*. For archaeostracans, drag reduction would have facilitated more efficient (and possibly faster) swimming, which is important for both pelagic and nektobenthic life modes, as assuming these arthropods were negatively buoyant, generation of lift would have also counteracted the negative buoyancy of the animals while swimming, again improving efficiency. This provides a possible explanation for the tentative link between tree path distance (number of nodes) from the benthic and burrowing *Arenosicaris* [24] and decreased drag and improved lift (Fig 6), as well as a pathway for pelagic taxa to evolve from benthic and nektobenthic ancestral stock, as repeatedly observed across a range of groups [43].

In this study, we focused on the role of the carapace outline, however this is just one of a number of morphological features that can improve swimming efficiency and speed in archaeostracans, or indeed any arthropod with an abdomen and specialised swimming appendages. Notably, the maximum lift, ranges of lift coefficients, and minimum drag coefficient for two Cambrian *Isoxys* carapaces (*I. longissimus* and *I. paradoxus;* [11]) are more hydrodynamically favourable than the best archaeostracans in this study. This suggests that for a stem-group arthropod without apparent appendage specialisation and without an abdomen, adaptation of the carapace may have played a more important role. For archaeostracans, there is no evidence that the abdomen could dorsally flex (and so a 'power stroke' as in *Nebalia* may not have been a major way of providing thrust [22]), however the abdomen may have still been important for escape reactions [22]. In addition to carapace morphology and the abdomen, thrust provided by specialised swimming appendages were likely also important for facilitating a pelagic mode of life in the group [20, 22], while the extremely thin carapaces (0.5–10 μm) relative to their length (c. 10–35 mm) [20] would have reduced the density of the animal and thus reduced energy expenditure for swimming.

## Conclusions

The carapace of pelagic caryocaridids generated broader ranges of lift coefficients and lower drag coefficients than the benthic *Arenosicaris inflata* at the same Reynolds number. However, some ceratiocaridids, *Ceratiocaris angusta* and *Ce. silicula*, display comparable hydrodynamic performance. Thus modification of the carapace shape was likely an important factor facilitating a pelagic mode of life in caryocaridids. Changes in carapace morphology, combined with improvement in swimming power through adaptations to appendages, as well as thinning of the carapace, were important for the explosion of caryocaridid diversity into the water column during the Ordovician.

## Supporting information

**S1 File. Outline_sources_and_ANSYS_results. xlsx an excel spreadsheet that includes the references and number of points sampled for each outline used in the analysis, and the ANSYS results for each outline.**
(XLSX)

**S2 File. CSV results a .zip folder containing a .csv file with the ANSYS results for each outline, formatted to be read into R.**
(ZIP)

**S3 File. R_code a .zip folder containing two .R files.** The first with the code used to convert the outlines from .jpg files into a format that can be read by ANSYS, and the second used for plotting the results.
(ZIP)

**S4 File. JPG_outlines a .zip folder containing the .jpg outlines for each taxon in this study.**
(ZIP)

## Acknowledgments

We thank the three anonymous reviewers and editor for their comments and feedback on the manuscript, and for carefully checking the code, data and supporting information.

## Author Contributions

**Conceptualization:** Stephen Pates.

**Data curation:** Stephen Pates, Yuan Xue.

**Formal analysis:** Stephen Pates, Yuan Xue.

**Investigation:** Stephen Pates, Yuan Xue.

**Methodology:** Stephen Pates, Yuan Xue.

**Project administration:** Stephen Pates.

**Resources:** Stephen Pates.

**Validation:** Stephen Pates, Yuan Xue.

**Visualization:** Stephen Pates.

**Writing – original draft:** Stephen Pates.

**Writing – review & editing:** Yuan Xue.

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
