## [Decision Letter · Decision Letter 0]

19 Feb 2024

PONE-D-23-26410Hydrodynamic performance of early Palaeozoic archaeostracan carapacesPLOS ONE

Dear Dr. Pates,

Thank you for submitting your manuscript to PLOS ONE. After careful consideration, we feel that it has merit but does not fully meet PLOS ONE’s publication criteria as it currently stands. Therefore, we invite you to submit a revised version of the manuscript that addresses the points raised during the review process.

We look forward to receiving your revised manuscript.

Kind regards,

Shuan-Hong Zhang, PhD

Academic Editor

PLOS ONE

Journal Requirements:

2. In your manuscript, please provide additional information regarding the specimens used in your study. Ensure that you have reported human remain specimen numbers and complete repository information, including museum name and geographic location. 

For more information on PLOS ONE's requirements for paleontology and archeology research, see https://journals.plos.org/plosone/s/submission-guidelines#loc-paleontology-and-archaeology-research.

Additional Editor Comments:

Dear Dr. Stephen Pates,

Thank you very much for your patience in waiting for reviewer’s comments for an extremely long time. I have now received two constructive and helpful reviews of your manuscript. As you can see from below, both reviewers believe the topic of your manuscript is interesting, but identify a number of critical issues in the current form of your manuscript.

Based on reviewer’s comments and my evaluation of your manuscript, I recommend reconsideration of your manuscript following major revisions. Please follow the comments and suggestions from reviewers (including PDF with comments by Reviewer 1#) and revise your manuscript carefully.

I am looking forward to receiving the revised version of your manuscript in the near future.

Sincerely,

Shuan-Hong Zhang

Reviewers' comments:

Reviewer's Responses to Questions

**Comments to the Author**

1. Is the manuscript technically sound, and do the data support the conclusions?

Reviewer #1: Partly

Reviewer #2: Partly

2. Has the statistical analysis been performed appropriately and rigorously? 

Reviewer #1: Yes

Reviewer #2: N/A

3. Have the authors made all data underlying the findings in their manuscript fully available?

Reviewer #1: No

Reviewer #2: Yes

4. Is the manuscript presented in an intelligible fashion and written in standard English?

Reviewer #1: Yes

Reviewer #2: Yes

5. Review Comments to the Author

Reviewer #1: Dear authors,

I found your research very interesting and of broad interest. However, there is, in my opinion a significant problem that, at the very least, needs to be discussed: you used 2D outlines from reconstructions published in several articles from the literature. There are two main issues with this. First, you cannot know if the reconstructions are made using comparable methods, so that species may not be comparable. Second, and worse, some species you used are not fully known in their outline and those reconstructions extrapolate the missing outline, that is the case for J. raymondi and C. angusta. For J. raymondi, we lacks at least 1/4th of the carapace anteriorly.

I think it is necessary to correct these issues, clearly state in the manuscript how many simulations were made, and what are the results for each species.

Sincerely

Reviewer #2: Manuscript PONE-D-23-26410 reports on the hydrodynamic performance of Cambro-Ordovician archaeostracan carapaces to discuss their lifestyle and ecological evolution across the Great Ordovician Biodiversification Event. The paper is overall well-written and well-illustrated, and represents an interesting contribution on the topic.

There is, however, an important issue with the antero-posterior orientation of the carapaces that absolutely needs to be fixed before this work can be considered for publication: most carapaces are oriented with the anterior region on the left, but Ceratiocaris silicula and Ceratiocaris angusta are represented antero-posteriorly inverted (i.e. with the anterior region on the right). The error probably comes from the erroneous use of the drawings in fig. 4 of Liu et al. 2022 (your ref. 21) that shows left valves for A and B (Pumilocaris granulosa & Pumilocaris salina) but right valves for Ceratiocaris silicula & Ceratiocaris angusta. I feel that there is also a similar problem of orientation for Arenosicaris inflata following fig. 11.1 of Collette & Hagadorn 2010 (your ref. 24).

Please re-run the analyses with the correct orientation. I guess, because the paper is short, that you could even use this “mistake” to further discuss the impact of such errors of orientation in CFD analyses, typically by comparing your current CFD results for the erroneous orientation (posterior to the left) with the results obtained for the correct orientation (anterior to the left).

I am also listing below additional minor suggestions/corrections (suggestions mostly towards giving a bit more explanations on the interpretation of the CFD data for paleontologists who are not familiar (yet!) with the approach):

- I would replace “early Palaeozoic” in the title with “Ordovician” for it be be more self-explainatory

- Ref. 22 “Nebalia bipes” should be italicized

- line 52, drop a bracket in “((25)”

- line 54 drop the coma in “using R (26), Momocs package (27)”

- line 61 You justify later (lines 64–66) the choice of scaling all models to the same length of 25mm, but considering that the size of caryocaridids was highly variable (and considering also that the paper is short) maybe also elaborate a little on the impact of size on the performed analyses (we can expect that some -if not most- of the readers might not be familiar with Computational Fluid Dynamics).

- lines 63-64 refers to Figure 2 while Figure 1 has not been called yet. Figure 1 is actually not called at all in the text. Refer to Figure 1 in the previous paragraph (“Carapace outlines”) or, and it might be better, add a Methods paragraph about “Taxonomic sampling” that refers to Figure 1 and justify the choice / reject of the taxa.

- line 88 paragraph “Mesh sensitivity”: because you have two different types of meshing presented in the Methods section, one for the generation of your 64-point carapace outline and the other for the CFD analyses, please remind here the one you are talking about

- line 93 “wereless” lacks a space.

- line 106 “were broadest the two Ceratiocaris species” lacks a work, for the two? In the two?

- Figures 4 & 5, for clearer visualization, would it be possible to use a color that is not part of the white-to-blue magnitude scale for the carapaces themselves (e.g. red or green), so that the highest pressures and velocities are more easily seen?

- lines 134–135 “These results indicate that the carapace was not well adapted for a pelagic mode of life.” It would be nice, for the neophyte, to explain a bit more why

- Fig. 6, it would be nice to find a way to identify the different taxa, I guess you could give each point a letter and indicate the taxon name corresponding to each letter in the figure caption.

- lines 158–160, the sentence “A functional comparison can be made to the elongate posterior spines are known in a likely pelagic Cambrian stem-group arthropod Isoxys longissimus.” reads weird, I guess a word is lacking or one must be replaced (or maybe “are” needs to be dropped?)

- line 310 “files(25)” lacks a space

- lines 313 & 318 “Arenosicaris inflata (a) Caryocaris acoitensis” lacks a come after (a)

- lines 314 & 319, drop the s for “0 degrees”

Finally, I want to really commend the authors for sharing their data and scripts! I have run the scripts and confirm that everything works. Nevertheless, the way OSF.io works requires renaming correctly the download .zip folders so that they match what’s needed for the R scripts. Please add comments to the R scripts to specify how the users need to save the downloaded data to make sure they get it working more easily. It is only about folder naming for script “Plottingresults_Caryocaridids.R”, but it’s much less straightforward for the other script “Outline_conversion.R”: the setwd path is not the same as the one for the other script (quite confusing) and I had to change line “CaryoCharacters<-as.vector(c('jpg','number','genus','species'))” into “CaryoCharacters<-as.vector(c('number','genus','species'))”, and line “lf <- list.files('JPG_noshortpostspines', full.names = TRUE)” into “lf <- list.files(, full.names = TRUE)” to get it working. Please comment the script in a more straightforward way for users who may not be “fluent” in R.

6. PLOS authors have the option to publish the peer review history of their article (what does this mean?). If published, this will include your full peer review and any attached files.

Reviewer #1: No

Reviewer #2: No

---

## [Author Response · Author response to Decision Letter 0]

29 Feb 2024

A full response to the reviewer comments has been uploaded in the file Response_to_reviewers.docx

---

## [Decision Letter · Decision Letter 1]

26 Apr 2024

PONE-D-23-26410R1Hydrodynamic performance of Ordovician archaeostracan carapacesPLOS ONE

Dear Dr. Pates,

Thank you for submitting your manuscript to PLOS ONE. After careful consideration, we feel that it has merit but does not fully meet PLOS ONE’s publication criteria as it currently stands. Therefore, we invite you to submit a revised version of the manuscript that addresses the points raised during the review process.

We look forward to receiving your revised manuscript.

Kind regards,

Shuan-Hong Zhang, PhD

Academic Editor

PLOS ONE

Journal Requirements:

Additional Editor Comments:

Dear Dr. Stephen Pates,

Thank you very much for your patience in waiting for reviewer’s comments for a long time. I have now received a constructive and helpful review from a new expert on your revised manuscript. Based on reviewer’s comments and my evaluation of your manuscript, I recommend reconsideration of your manuscript following minor revisions. Please follow the comments and suggestions from reviewer and revise your manuscript carefully.

I am looking forward to receiving the revised version of your manuscript in the near future.

Sincerely,

Shuan-Hong Zhang

Reviewers' comments:

Reviewer's Responses to Questions

**Comments to the Author**

1. If the authors have adequately addressed your comments raised in a previous round of review and you feel that this manuscript is now acceptable for publication, you may indicate that here to bypass the “Comments to the Author” section, enter your conflict of interest statement in the “Confidential to Editor” section, and submit your "Accept" recommendation.

Reviewer #3: All comments have been addressed

2. Is the manuscript technically sound, and do the data support the conclusions?

Reviewer #3: Yes

3. Has the statistical analysis been performed appropriately and rigorously? 

Reviewer #3: Yes

4. Have the authors made all data underlying the findings in their manuscript fully available?

Reviewer #3: Yes

5. Is the manuscript presented in an intelligible fashion and written in standard English?

Reviewer #3: Yes

6. Review Comments to the Author

Reviewer #3: Review: Hydrodynamic performance of Ordovician archaeostracan carapaces

General Comments

Pates and Xue provide an interesting article about the hydrodynamic properties of Ordovician archaeostracans. The article is sound and aligns with the standards of the field. The article has already been reviewed by two independent researchers besides this review. I believe that the authors have succesfully identified the main problems of their study and addressed them. I recommend publication after this round of review.

Evaluation of the comments addressed to previous reviewers

Reviewer 1: Reviewer 1 brings good points regarding data accuracy and about the sources of the outlines, as well as other helpful comments, which I think have adequately been corrected by the authors. Here, I review some of their main points.

1) The main problem highlighted by Reviewer 1 is the use of published 2D outlines from the literatures.

The authors have provided a new section (Taxonomic sampling) reviewing all outlines used and their origins, which in my opinion, would suffice. Regarding the use of 2D outlines, from my perspective:

Using outlines from the literature is a standard practice in geometric morphometric analyses of palaeozoic invertebrates, especially when outlines are as simple as carapace valves (e.g., Caron & Moysiuk, 2021RSOS). Ideally, the authors would have examined the material themselves using state-of-the-art imaging. However, this requires an important investment in time and economic resources, including multiple international trips, access to different collections (...) to acquire a complete dataset.

Using previously described outlines is an adequate solution to this problem in tandem with a careful examination of any other type of evidence (most commonly, images provided by the different institutions or included in their respective publications). Using images alone from previous publications can also introduce biases, as their quality in older publications is often low. Although new technologies (e.g., cross-polarizing photography) have improved our examination of fossil material, past researchers often excelled at diagrammatic reconstructions, and, unless there are major differences between their reconstructions and the images available, these reconstructions can normally be used as reliable evidence.

The study focuses on interspecific differences and thus, even if there are small changes between the real carapace shape and the reconstruction/image, these differences do not impact heavily interspecific differences, as the authors comment in their main text.

2) Wrong reconstructions for J. raymondi and C. angusta.

Reviewer 1 correctly identifies this issue. The authors have correctly added a note in their main text. Given that these species are not relevant in the discussion, I believe is adequate to include them on account of including as many Ordovician archeostracans as possible.

3) 3D analyses.

I agree with the authors that a 3D analysis, while interesting, would also introduce additional variables to consider. Carapace valves are usually found in isolation and two-dimensionally compressed. 3D analyses of other palaeozoic arthropods like trilobites can only be easily peformed because they are, in nature, three-dimensional fossils. In order to perform a 3D analysis, the authors must reconstruct two main features: the attachment between both valves (i.e., the closure of the carapace) as well as the three-dimensionality of the carapace valve (i.e., whether it is bulging or flat). These two factors could be extrapolated using multiple specimens of the same species showing compression artefacts and different orientations of the carapace (in different views, for example), and taking into account the mode of preservation. This is already quite difficult for a single species, for which this material may not be present.

Furthermore, as the authors point out, in order to create a hydrodynamic model, a 2D shape is suitable. To my knowledge, there is no specific criteria data must follow to create an hydrodynamic model. There have already been similar studies using the same methods and 2D shapes (e.g., Pates et al., 2021 ProcB; Meyers & Msomi, 2021, Materialstoday) even in engineering journals (e.g., Mohamed et al., 2021. J.W.E.I.A). Of course, the more data available, the more accurate the hydrodynamic model is, including three-dimensional models, effect of the legs and tail, size, type of cuticle... These features may be easily accessible for extant taxa, but are difficult to obtain in the fossil record, and thus, a simple 2D approach is appropriate as long as the discussion and conclusion focus on broad patterns.

Reviewer 2: Reviewer 2 brings valuable comments that, in my opinion, have correctly been assessed by the authors. The main issue brought by the reviewer was the incorret orientation of two species of the dataset, which has been corrected accordingly.

Additional reviews

I think the abstract does not completely capture the results of the paper. To me, the main points are that there is a general trend across archeostracan evolution towards lowering drag and improving lift, which is quite clear in the origin of caryocaridids, which could be related to their change into a pelagic lifestyle. It is good to acknowlege further adaptations, but the importance of the appendages or thinning of the carapace are not widely covered in the main text. I think at least a mention to drag and lift is necessary, and I encourage the authors to decide on their best way to highlight their other results.

Smaller comments

34- delete extra space between of and arthropods

39- can the authors provide a reference or reconsider the word "zoonekton"? I don't think this a very common term; a quick search reveals almost no previous references using it.

197- what does "wake" mean in this sentence?

210- change "a" to "at"

210-211- I think that the sentence "taxa at a higher number of nodes" should be changed. Although I understand what the authors mean, it is not very accurate, as the number of nodes depend on the number of diversification events across the tree. I think what the authors mean is that in general, caryocaridids, have lower drag coefficient and higher lift:drag ratio than ceratiocarids, which also have lower drag and higher lift than Aresinocaris.

213- please specify the correlation, as the reader may think is between drag and lift. I think what the authors mean is that there may be a trend towards lower drag and higher lift towards the origin of caryocaridids but cannot test it with (e.g.) phylogenetic tracing.

233- Ce. angusta (italics)

235- delete ";" after lethiersi

241- Thus,

260- Again, the use of "the node above" is a little bit strange from a phylogenetic perspective. In this case, for example, I think it is easier to state that the morphology probably appeared at the common ancestor of all Ordovician archeostracans, and was secondarily lost in Pumilocaris.

275-better state "This suggests that for this stem-group arthropod without apparent appendage specialization and without abdomen, adaptations of the carapace played a more important carapace role". This is because "were not possible" assumes that their stem-group position limits their potential morphological adaptations. This is not something we can easily assume, especially when similar taxa like Occacaris have an abdomen and that new information is increasingly revealing new adaptations regarding the legs of Isoxys (e.g., the endites recorvered by Zhang et al., 2021).

280- Maybe the authors could separate this into different sentences, as it is a little bit unclear to me when written as one singular sentence. By elongating the body, do the authors mean that the animal would stretch its abdomen? If that is the case, how does it change the separation of the flow? Would swimming speed necessary to achieve flow separation be reduced, then? Is there any citation or further evidence (e.g, carapace shape) the authors could use to back up this claim?

References: I have similarly experienced problems with Mendeley's citations. My recommendation to the authors is to edit the PLOS style in Mendeley under a different name and make sure to always use the same account and preferably, device. Mendeley often fails to update properly across different devices, so changing the document from one to another device can lead to errors.

323: bioturbation: an updated account.

325: Isoxys (italics)

328: ecosystem: evidence

333: zooplankton: bradoriid

342: (GOBE): the palaeocological

347: trilobites: the invasion

355: Microparia speciosa (italics=

387: Dimorphism of bivalved arthropod

405: Momocs: outline analysis

407: Swimming soeed and oxygen consupmtion in the bathypelagic mysid

416: Origins, evolution and diversification of zooplankton.

7. PLOS authors have the option to publish the peer review history of their article (what does this mean?). If published, this will include your full peer review and any attached files.

Reviewer #3: No

---

## [Author Response · Author response to Decision Letter 1]

8 May 2024

We have uploaded a document with our response to the reviewer.

---

## [Editor Report · Decision Letter 2]

15 May 2024

Hydrodynamic performance of Ordovician archaeostracan carapaces

PONE-D-23-26410R2

Dear Dr. Pates,

We’re pleased to inform you that your manuscript has been judged scientifically suitable for publication and will be formally accepted for publication once it meets all outstanding technical requirements.

Kind regards,

Shuan-Hong Zhang, PhD

Academic Editor

PLOS ONE

Additional Editor Comments (optional):

Dear Dr. Stephen Pates,

Thank you very much for revising the manuscript following the reviewer’s suggestions. I am pleased to inform you that your manuscript has been accepted for publication in PLOS one.

Best wishes,

Shuan-Hong Zhang
---

## [Editor Report · Acceptance letter]

21 May 2024

PONE-D-23-26410R2 

PLOS ONE

Dear Dr. Pates, 

I'm pleased to inform you that your manuscript has been deemed suitable for publication in PLOS ONE. Congratulations! Your manuscript is now being handed over to our production team.

Kind regards, 

on behalf of

Dr. Shuan-Hong Zhang 

Academic Editor

PLOS ONE